# Identifying the distinct roles of dual dopants in stabilizing the platinum-nickel nanowire catalyst for durable fuel cell

Lei Gao[1,5], Tulai Sun[2,5], Xuli Chen[1,5], Zhilong Yang[1], Mengfan Li[1], Wenchuan Lai [1], Wenhua Zhang [3], Quan Yuan [1] & Hongwen Huang [1,4] ✉

Stabilizing active PtNi alloy catalyst toward oxygen reduction reaction is essential for fuel cell. Doping of specific metals is an empirical strategy, however, the atomistic insight into how dopant boosts the stability of PtNi catalyst still remains elusive. Here, with typical examples of Mo and Au dopants, we identify the distinct roles of Mo and Au in stabilizing PtNi nanowires catalysts. Specifically, due to the stronger interaction between atomic orbital for Ni-Mo and Pt-Au, the Mo dopant mainly suppresses the outward diffusion of Ni atoms while the Au dopant contributes to the stabilization of surface Pt overlayer. Inspired by this atomistic understanding, we rationally construct the PtNiMoAu nanowires by integrating the different functions of Mo and Au into one entity. Such catalyst assembled in fuel cell cathode thus presents both remarkable activity and durability, even surpassing the United States Department of Energy technical targets for 2025.

Proton exchange membrane fuel cells (PEMFCs) have shown great potential as power sources for electric vehicle applications owing to their high energy conversion efficiency and environmental benign characteristics[1–4]. However, the widespread uptake of such PEMFCs in automotive vehicles is largely impeded at present because a high loading amount of precious Pt on the cathode (-0.4 mg cm$^{-2}$) is needed to promote the sluggish kinetics of oxygen reduction reaction (ORR) and compensate the insufficient catalyst durability[5–8]. Alloying Pt with a transition metal, such as Fe, Co, Ni, and Cu, has been identified as an effective way to boost the catalytic activity toward ORR through the advantageous ligand and strain effects[9–14], which has thus been regarded as the most promising solution to reduce the excessive Pt usage. As a convincing example, the spherical PtCo alloy nanocatalysts are adopted in the Toyota Mirai fuel cell vehicle at present[4]. Unfortunately, the long-term durability of these active PtM (M = transition metals) alloy catalysts still remains a serious problem, which can be generally ascribed to the rapid leaching of the reactive transition metal

under detrimentally corrosive ORR conditions and "start-up/shut-down" process (the applied potential is generally over 1.0 V)[15–17]. Under this context, how to stabilize the active PtM alloy catalyst for long-term fuel cell operation has become the most concerned research topic for the industrial application of PEMFCs.

Doping a specific element, with notable examples of Mo, Au, and Rh, into active PtM alloy catalysts and metal oxide support has been discovered as a straightforward and potential strategy to enhance the catalytic durability of PtM alloy catalysts for ORR[18–25]. Mechanistically, the prior studies suggested that these dopants may stabilize the alloy catalysts via suppressing the surface Pt migration and/or mitigating the outward diffusion of reactive transition metal[18]. However, it needs to be pointed out that these two main reasons (i.e., surface Pt migration and outward diffusion of reactive transition metal) are essentially entangled[18,26–28], making great difficulty to gain a clear mechanistic understanding. Moreover, those arguments on the stabilization mechanism of dopants were mostly established via theoretical

[1]College of Materials Science and Engineering, State Key Laboratory of Chemo/Biosensing and Chemometrics, Hunan University, Changsha, Hunan, PR China. [2]Center for Electron Microscopy, State Key Laboratory Breeding Base of Green Chemistry Synthesis Technology and College of Chemical Engineering, Zhejiang University of Technology, Hangzhou, Zhejiang, PR China. [3]Department of Chemical Physics, University of Science and Technology of China, Hefei, Anhui, PR China. [4]Shenzhen Research Institute of Hunan University, Shenzhen, Guangdong, PR China. [5]These authors contributed equally: Lei Gao, Tulai Sun, Xuli Chen. ✉ e-mail: huanghw@hnu.edu.cn

simulations, probably because of the challenges in constructing the model catalysts, more or less leading to the discrepancy with a realistic situation. As a result, the atomistic understanding of the role of dopants that is concluded from a comprehensive combining of convincing experiments and theoretical computations is still lacking presently, which indeed greatly limits the rational design of active and stable PtM alloy catalysts for practical fuel cells.

Because Mo and Au elements are resistant to corrosion under acidic ORR, these elements offer ideal cases for the stability study. As a result, we systematically investigate the roles of Mo and Au dopants in stabilizing PtNi alloy catalysts based on the well-defined PtNi-based nanowires (NWs) model catalysts. Combining ex-situ experimental characterizations and density of functional theory (DFT) calculations, the distinct roles of Mo and Au in stabilizing PtNi NWs catalysts are identified. We find that the Mo dopant plays a major role in suppressing the outward diffusion of Ni atoms and the Au dopant mainly stabilizes the surface Pt overlayer. Based on this atomistic understanding, PtNiMoAu NWs are rationally designed and constructed by integrating the different functions of Mo and Au into PtNi NWs. As expected, the as-synthesized PtNiMoAu NWs catalysts present an unprecedented activity and stability toward ORR, with a 16.2% loss in its mass activity (MA) after 80,000 (80 K) cycles of durability test. When assembling the PtNiMoAu NWs catalysts into the fuel cell cathode, a high MA retention of 77.4% ($H_2$-$O_2$, 0.9 $V_{iR-free}$) and a low voltage loss of 25 mV ($H_2$-Air, 0.8 A cm$^{-2}$) after 30 K cycles of durability test are output, proving the highly durable fuel cell performance.

## Results

### Synthesis and characterizations of PtNi-based model catalysts

To explore the effect of Mo and Au dopants on the ORR performance of the PtNi catalyst, we synthesized a series of PtNi-based NWs, containing PtNi NWs, PtNiMo NWs, and PtNiAu NWs with the same diameter, as model catalysts by slightly adjusting the synthetic method (Supplementary Table 1). The detailed structure and composition of the three NWs were carefully analyzed. The low-magnification transmission electron microscopy (TEM) images in Supplementary Fig. 1a–c shows the successful preparation of PtNi NWs, PtNiMo NWs, and PtNiAu NWs in high purity and uniformity. The average diameters of these three NWs are all about 1.2 nm, which were determined by counting 100 NWs (Supplementary Fig. 1d–f). To acquire the atomic-level structural information of three NWs, we captured the atomically resolved high-angle annular dark-field scanning transmission electron microscopy (HAADF-STEM) image of a single NW (Fig. 1a–c). Specifically, for all these three NWs, a typical NW is made up of about six atomic layers, agreeing well with the average diameter. And the interplanar spacings of {111} planes for three NWs are almost the same, suggesting that the negligible strain is introduced with the doping of Mo or Au atoms into PtNi NWs. As shown in Fig. 1d, the X-ray diffraction (XRD) pattern of three NWs presented the same features in terms of peak number and peak position, being consistent with the data from its interplanar spacing. Figure. 1a–c and Supplementary Fig. 2 show the energy-dispersive X-ray spectroscopy (EDS) elemental mapping and EDS line-scanning profile, clearly demonstrating the homogeneous

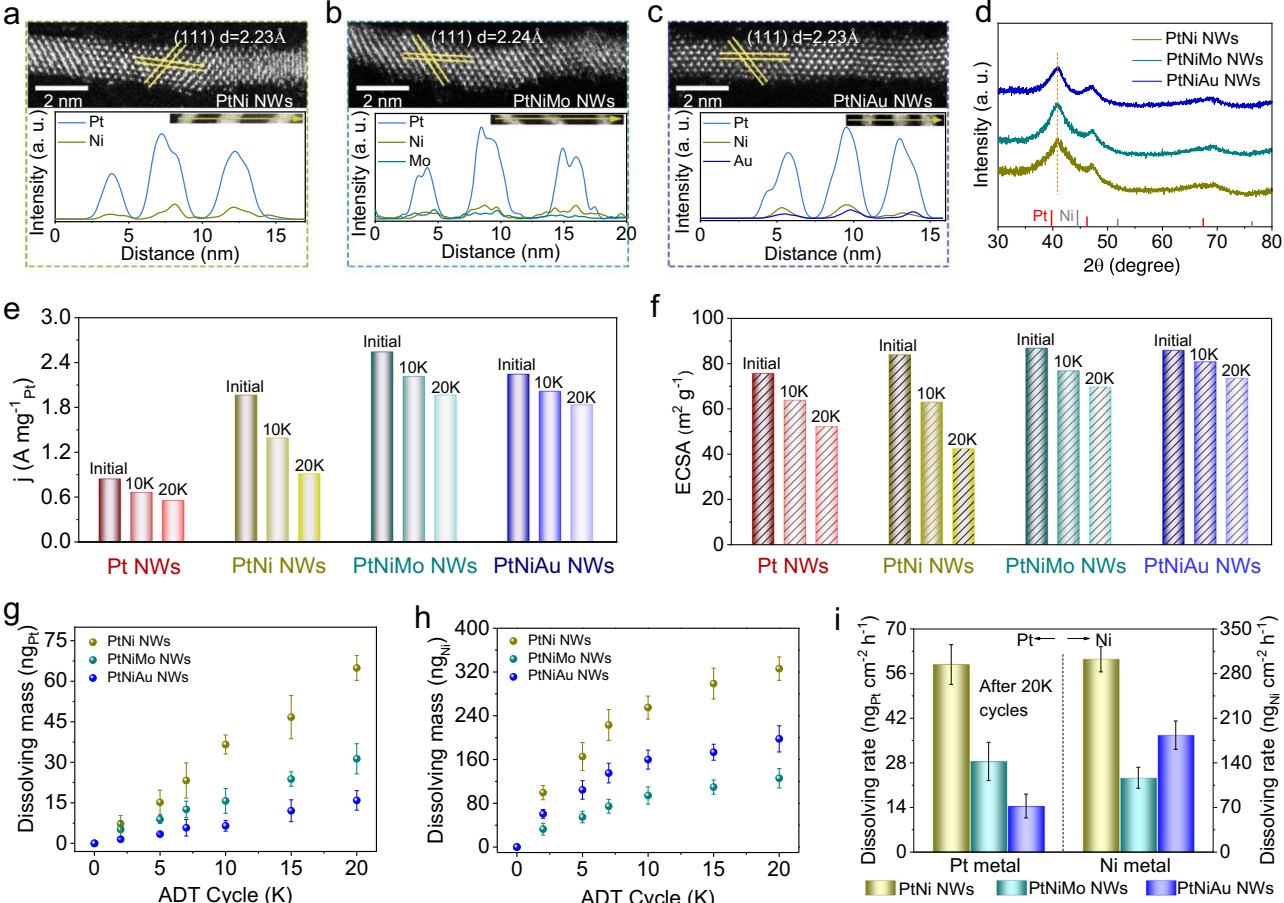

**Fig. 1 | Structural characterizations and ORR performance.** HAADF-STEM images and EDS EDS line-scanning profiles of **a** PtNi NWs, **b** PtNiMo NWs, and **c** PtNiAu NWs. **d** XRD patterns of PtNi-bassed NWs. **e** MA and **f** ECSAs for different NWs/C catalysts after different cycles. Dissolving mass of **g** Pt and **h** Ni metals for PtNi-based NWs/C catalysts after different cycles. **i** Dissolving rate of Pt and Ni metals for PtNi-based NWs/C catalysts after 20 K cycles. All dissolving mass and dissolving rate values are means with error bars (standard deviations) from three replicates.

distribution of Pt/Ni in PtNi NWs, Pt/Ni/Mo in PtNiMo NWs, and Pt/Ni/Au in PtNiAu NWs. According to the inductively coupled plasma mass spectrometry (ICP-MS), the atomic ratio of Pt/Ni in PtNi NWs, Pt/Ni/Mo in PtNiMo NWs, and Pt/Ni/Au in PtNiAu NWs were determined to be 3.03:1.00, 2.99:1.00:0.10, and 3.01:1.00:0.11, indicting the similar atomic ratio of Pt:Ni in three NWs. Taken together, the as-prepared PtNi-based NWs possess almost identical morphological structure, diameter, ratio of Pt/Ni, and interplanar spacing, thus offering the ideal platform to study the effects of Mo and Au dopants on the catalytic stability of PtNi NWs toward ORR.

## Atomistic Insights into the Roles of Mo and Au Dopants for Catalytic Stability

Prior to ORR measurements, three PtNi-based NWs, as well as the Pt NWs using as the reference catalyst (Supplementary Fig. 3), were first loaded on Vulcan XC-7 carbon to prepare the carbon-supported NWs (NWs/C) catalysts (Supplementary Fig. 4). The cyclic voltammograms (CVs) and positive-going polarization curves of all NWs/C catalysts were further recorded to obtain the initial electrochemically active surface areas (ECSAs) and MA, respectively (Supplementary Fig. 5 and Fig. 1e, f). Clearly, the MA of PtNi NWs/C (1.96 A mg$^{-1}_{Pt}$) catalyst displayed an obvious increase with respect to the Pt NWs/C (0.84 A mg$^{-1}_{Pt}$), in line with the prior studies[29]. When the Mo and Au dopants are introduced, the MA could be further increased to 2.54 A mg$^{-1}_{Pt}$ (PtNiMo NWs/C) and 2.24 A mg$^{-1}_{Pt}$ (PtNiAu NWs/C), respectively. The long-term catalytic stability for these three PtNi-based NWs catalysts was then assessed via an accelerated durability test (ADT) between 0.6 and 1.0 V versus reversible hydrogen electrode (V$_{RHE}$) in O$_2$-saturated 0.1 M HClO$_4$ at room temperature. The CVs curves and polarization curves of NWs catalysts after certain cycles of ADT were then recorded, where the corresponding specific activity (SA), ECSAs, and MA were derived (Supplementary Figs. 6 and 7). The results demonstrate that both the Mo and Au dopants could significantly enhance the catalytic stability of PtNi NWs/C catalyst for ORR. Specifically, the PtNiMo NWs/C and PtNiAu NWs/C only show a drop of 22.8% and 18.3% in MA after 20 K cycles of ADT, respectively, contrasting with the big decrease of 53.5% for PtNi NWs/C catalyst (Fig. 1e). The changes in structure for three PtNi-based catalysts after 20 K cycles of ADT were further examined to support the observations on durability trend. Self-consistently, the PtNiMo NWs/C and PtNiAu NWs/C catalysts both present smaller changes in ECSAs and NWs diameters after ADT measurements relative to PtNi NWs/C catalyst (Fig. 1f, Supplementary Figs. 8 and 9). These results together manifest that both the Mo and Au dopants could improve the ORR performance of the PtNi NWs/C catalyst (Supplementary Table 2).

The in-depth insights into how Mo and Au dopants stabilize the PtNi catalyst for ORR operation were further explored. In principle, the Pt-based ORR catalysts degradation can be traced back to the following four possible reasons: (i) degradation of carbon support; (ii) dissolution of reactive metal components, e.g. the Pt and Ni for PtNi catalyst; (iii) detachment of catalyst nanoparticles; (iV) loss of active surface area induced by electrochemical Ostwald ripening and nanoparticle coalesce[30,31]. Because our PtNi-based model catalysts have almost identical structures and the same carbon support, we can safely exclude the carbon degradation and catalyst detachment as the reasons for the different stability performance of PtNi-based model catalysts. Besides, it should be noted that the dissolution of reactive metal components can trigger the electrochemical Ostwald ripening and nanoparticle coalesce[32]. In this case, we could reasonably assume that the dissolution of Pt and Ni elements should be the main cause for the PtNi-based catalysts degradation under long-term operation[33,34]. Consequently, we systematically detected the dissolution process of Pt and Ni elements for these catalysts during ADT measurements by a stationary rotating disk electrode coupled with an ex-situ ICP-MS. Fig. 1g, h shows the leaching amounts of Pt and Ni in the electrolyte after cyclic

testing at intervals, showing the continuous leaching of Pt and Ni for each catalyst. Intriguingly, it is found that the Pt dissolution is maximally suppressed when the Au dopant is introduced, as indicated by the lowest dissolution rate of Pt in PtNiAu after 20 K cycles (Fig. 1i). The difference is that the Ni dissolution is suppressed most when the Mo dopant is adopted. These experimental evidences definitely indicate that the Mo and Au dopants play distinct roles in improving the stability of PtNi NWs/C catalyst. Specifically, the Mo dopant mainly contributes to the Ni stabilization while the Au dopant dominantly restrains the Pt dissolution.

To further verify the distinct roles of Mo and Au dopants, the DFT calculations were performed. In light of the corresponding structural and compositional parameters, we first constructed the PtNiAu (111) slab, PtNiMo (111) slab, PtNi (111) slab (Supplementary Fig. 10) as the models to represent the PtNiAu NWs/C, PtNiMo NWs/C, and PtNi NWs/C catalysts, respectively. Since the vacancy formation energies (E$_{vac}$) of Pt and Ni atoms could reflect the tendency of Pt atoms dissolution and Ni atoms leaching[35,36], the E$_{vac}$ of Pt and Ni atoms for PtNi-based (111) slabs were calculated (Supplementary Fig. 11). In comparison with PtNi (111) slab, both PtNiMo (111) slab and PtNiAu (111) slab present increased E$_{vac}$ of Pt and Ni atoms, self-supporting the mitigated dissolution of Pt and Ni for PtNiMo NWs/C and PtNiAu NWs/C catalysts. We further found that the E$_{vac}$ of Pt atoms for the PtNiAu (111) slab and E$_{vac}$ of Ni atoms for PtNiMo (111) slab present the largest increment with respect to the PtNi-based slab, respectively (Fig. 2a). The results thus confirm that the Mo and Au dopants could separately mitigate the leaching of Ni and Pt atoms, in accord with experimental observations from the ICP-MS. Besides the E$_{vac}$, the diffusion energy barrier of Ni is also crucial for Ni leaching. We thus calculated the Ni diffusion energy diagrams for PtNi-based slabs (Fig. 2b, Supplementary Fig. 12). It could be seen that the diffusion energy barrier of Ni for PtNiMo (111) slab (0.68 eV) was the largest, followed by PtNiAu (111) slab (0.57 eV) and PtNi (111) slab (0.40 eV), indicating that Mo dopant could obviously inhibit the outward diffusion of bulk Ni atoms (Fig. 2b). All these preceding results together confirm that, in improving the durability of PtNi NWs, the Mo dopant mainly increases E$_{vac}$ and diffusion energy of Ni atoms, while the Au dopant mainly increases E$_{vac}$ of Pt atoms.

We further attempted to understand how the Mo and Au dopants play distinct roles in stabilizing Ni and Pt, which should be, in principle, originated from the different interaction strengths between dopants and Pt/Ni[18,26]. To explore the interaction strength for the atomic pair of Mo-Pt, Au-Pt, Mo-Ni, and Au-Ni, the partial density of states (PDOS) of PtNiMo (111) and PtNiAu (111) slabs were analyzed (Fig. 2c, d). In general, the degree of overlaps between d-orbitals could reflect bonding strength between different metals. The results show that the degree of overlap between Pt d-orbital and Au d-orbital is larger than that of Pt d-orbital and Mo d-orbital, indicating the reinforced interaction between Pt and Au. Meanwhile, the stronger interaction between Mo d-orbital and Ni d-orbital is found relative to that between Au d-orbital and Ni d-orbital. These insights into the interaction between dopants and Pt/Ni atoms reasonably underpin the distinct roles of Mo and Au dopants in stabilizing the PtNi catalyst.

Besides, the improved activities of PtNiMo NWs/C and PtNiAu NWs/C catalysts were also rationalized by DFT calculations (Supplementary Fig. 13). Supplementary Fig. 14a, b shows the Gibbs free energies of reaction intermediates for the ORR on different slabs. The results testify that the overpotential on Pt (111), PtNi (111), PtNiMo (111), and PtNiAu (111) slabs follows the trend of PtNiMo (111) < PtNiAu (111) < PtNi (111) < Pt (111), well-consistent with our experimental observations on the activity trend. Relative to PtNi NWs/C catalyst, we could conclude that the weakened adsorption of oxygenated species on PtNiMo NWs/C and PtNiAu NWs/C catalysts arising from the ligand effect is the intrinsic cause for the improved activity, which is further supported by the downshifted d-band center on PtNiMo (111) and PtNiAu (111) slabs (Supplementary Fig. 14c)[35,37,38].

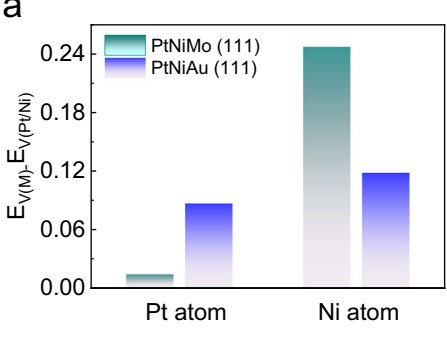

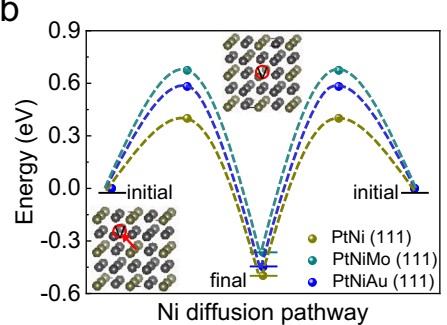

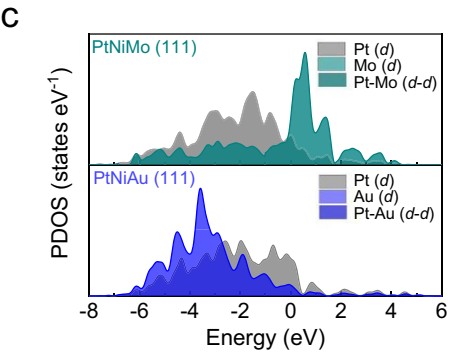

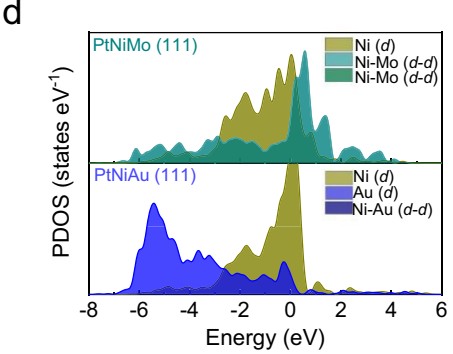

**Fig. 2 | Stability of catalysts based on DFT calculations. a** The change of Pt and Ni vacancy formation energies ($E_{V(Pt)}$) for PtNiMo (111) and PtNiAu (111) slabs relation to PtNi slab. **b** Ni diffusion pathway energy diagram for different slabs. The inset showed the proposed Ni diffusion pathway in the bulk alloy. **c**, **d** The PDOSs. Site-dependent PDOSs of Pt-5$d$, Ni-3$d$, Mo-3$d$ and Au-3$d$ on PtNiMo (111) and PtNiAu (111) slabs.

## Rational design and synthesis of PtNiMoAu NWs

Since Au and Mo dopants could respectively stabilize the surface Pt overlayer and suppress the leaching of Ni atoms, it is reasonably hypothesized that integrating these two dopants into PtNi would further improve the ORR stability. The hypothesis is further verified by the DFT calculations, as indicated by the largest $E_{vac}$ of Pt and Ni atoms, as well as the highest diffusion energy barrier (0.76 eV) of Ni atoms, on PtNiMoAu (111) slab (Supplementary Figs. 10–12 and 15a). The strong coupling degrees between Mo, Pt, and Au $d$ orbitals, as well as Mo, Ni, and Au $d$ orbitals may account for the increased $E_{vac}$ and diffusion energy barrier (Supplementary Fig. 15b, c). Moreover, it is predicted that the PtNiMoAu would present the enhanced ORR activity, as indicated by the lowest overpotential and downshifted $d$-band center of the PtNiMoAu (111) slab (Supplementary Fig. 14). Inspired by this conceptual design, we subsequently synthesized PtNiMoAu NWs catalysts by using a modified synthetic procedure (see Methods for detail).

Figure 3a,b shows low-magnification TEM and STEM images of the as-synthesized PtNiMoAu NWs, respectively. The NWs with a diameter of $1.1 \pm 0.3$ nm and length of $62 \pm 21$ nm are identified by statistic counting (Fig. 3c). The atomic-level structural information was analyzed based on the HAADF-STEM image of a single NW (Fig. 3d). Specifically, the well-defined lattice spacing can be assigned to the {200} and {111} planes, matching with the fast Fourier transform (FFT) pattern in the inset. From the distinguished lattice planes, the grown direction of <110> orientation could be deduced. The EDS elemental mapping (Fig. 3e) and line-scanning profile (Fig. 3f) both indicate the homogeneous distribution of Pt, Ni, Mo, and Au elements throughout a single NW. The Pt:Ni:Mo:Au atomic ratio was estimated to be 3.02:1.00:0.11:0.13 by ICP-MS, very close to the value (3.10:1.00:0.15:0.14) determined by the XPS spectrum (Supplementary Fig. 16). These structural characterizations prove the successful formation of ultrathin PtNiMoAu tetrametallic NWs.

The chemical states for the constituent elements of PtNi NWs and PtNiMoAu NWs were then analyzed by XPS. Compared with the Pt 4$f$ of PtNi NWs, the decreased binding energy for Pt 4$f$ of PtNiMoAu NWs implies more electrons accumulate around Pt in PtNiMoAu NWs (Fig. 3g). To examine the local coordination and electronic structures of the PtNi NWs and PtNiMoAu NWs, we further employed X-ray absorption spectroscopy (XAS), in comparison with a bulk Pt foil and PtO$_2$. Figure 3h shows the X-ray absorption near-edge structure (XANES) spectra of Pt L$_3$-edge. Agreeing well with XPS results and Bader analysis (Supplementary Table 3), the Pt white line of PtNiMoAu NWs is found to possess lower intensity and be close to the metallic state (Pt foil) when compared to the PtNi NWs. These results manifest that, after introducing the Mo and Au atoms, Pt atoms in PtNiMoAu NWs could gain more electrons. The local structure was further derived from the Fourier transform of the phase-corrected extended X-ray absorption fine structure (EXAFS). Fig. 3i and Supplementary Fig. 17 show the fitted R-space and K-space data of the Pt edge for PtNiMoAu NWs and PtNi NWs. Clearly, the first-shell Pt-Pt length of PtNi NWs and PtNiMoAu NWs was obviously shorter than that of Pt/C, attributing to the smaller radius of Ni/Mo atoms. Besides, the PtNiMoAu NWs present a similar Pt-Pt coordination number ($5.5 \pm 0.4$) with that of PtNi NWs ($6.3 \pm 0.4$), due to the similar NWs diameter (Supplementary Table 4). These results together demonstrate that the Mo and Au dopants could redistribute the electrons and induce the compressive strain, which are believed as the structural foundations for the improved ORR performance as predicted.

## ORR Performance of PtNiMoAu NWs

To verify the validity of our rational design for a practical catalyst, we further assessed the ORR performance of PtNiMoAu NWs/C catalyst by recording the CVs and polarization curves in 0.1 M HClO$_4$ solution (Supplementary Fig. 18 and 19). Specifically, the PtNiMoAu NWs/C catalyst exhibits specific activity (SA) and MA of 3.13 mA cm$^{-2}$ and

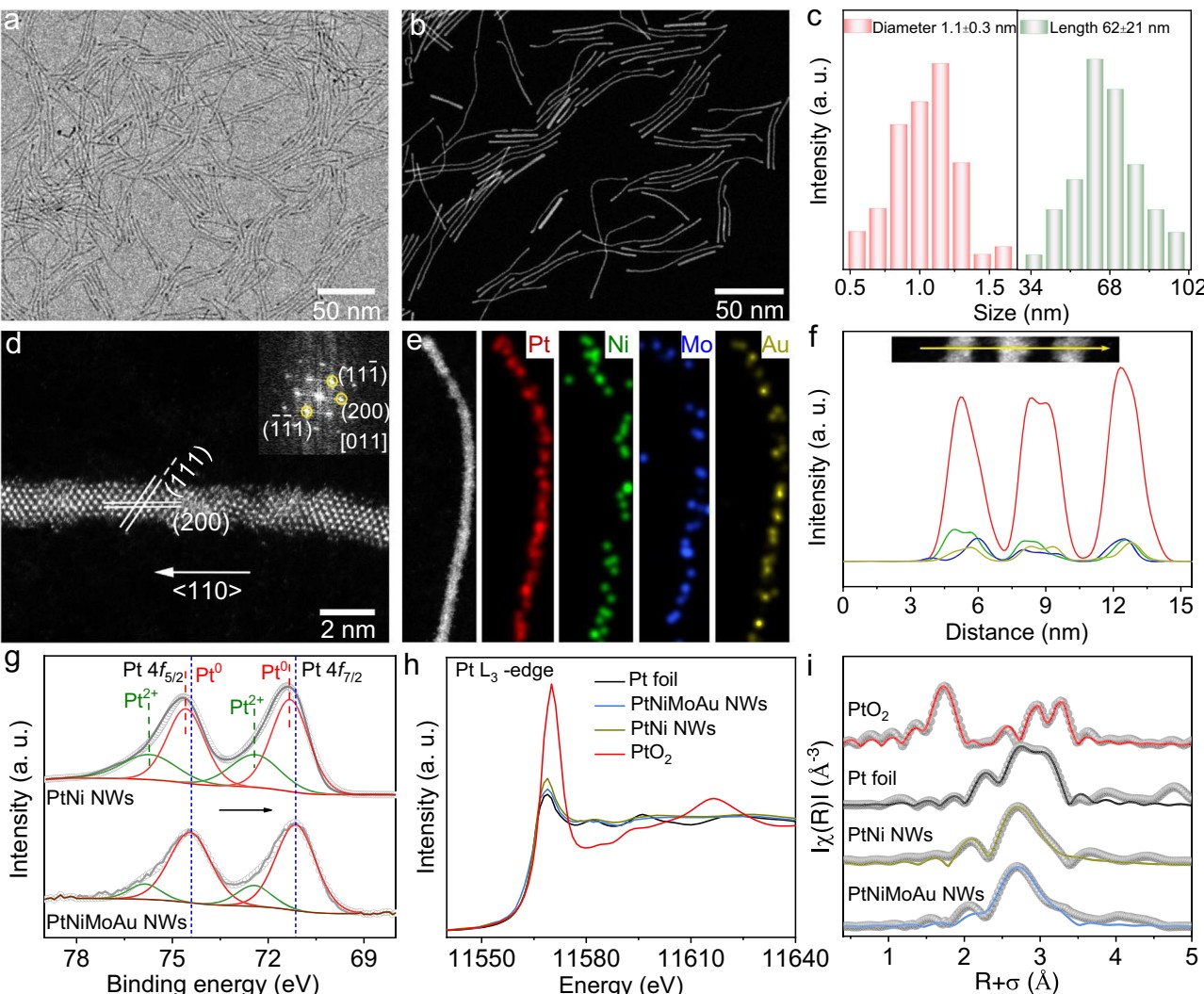

**Fig. 3 | Structural and compositional characterizations of ultrathin PtNiMoAu tetrametallic NWs. a** Low-magnification TEM image. **b** Low-magnification STEM image. **c** Histogram of diameter and length distributions. **d** Atomically resolved HAADF-STEM image. The inset shows the corresponding FFT pattern. **e** STEM-EDS elemental mapping profiles. **f** EDS line-scanning profiles. **g** High-resolution XPS spectra of Pt $4f$ for PtNi NWs and PtNiMoAu NWs. **h** Pt $L_3$-edge XANES spectra and **i** the Fourier transforms of EXAFS spectra for different samples.

2.89 A mg$^{-1}_{Pt}$ at 0.9 V$_{RHE}$, much higher than those of Pt/C (0.27 mA cm$^{-2}$ and 0.18 A mg$^{-1}_{Pt}$), PtNi NWs/C (2.34 mA cm$^{-2}$ and 1.96 A mg$^{-1}_{Pt}$), PtNiMo NWs/C (2.93 mA cm$^{-2}$ and 2.54 A mg$^{-1}_{Pt}$) and PtNiAu NWs/C catalysts (2.61 mA cm$^{-2}$ and 2.24 A mg$^{-1}_{Pt}$) (Fig. 4a and Supplementary Fig. 19). These results prove that the co-doping of Mo and Au elements into PtNi NWs evidently boost the ORR activity, which is in accordance with DFT prediction.

The long-term stability of PtNiMoAu NWs/C catalyst was further evaluated by ADT. Figure 4b shows the polarization curves of PtNi-MoAu NWs/C catalyst after different cycles of ADT. Remarkably, the ECSAs (86.1 m$^2$ g$^{-1}_{Pt}$) and MA (2.42 A mg$^{-1}_{Pt}$) of PtNiMoAu/C catalyst only displayed a drop of 6.7% and 16.2% after 80 K cycles (Fig. 4c and Supplementary Fig. 20), respectively. Of note, such excellent long-term stability of PtNiMoAu NWs surpasses the recently-reported Pt-based ORR electrocatalysts at different cycles of ADT (Fig. 4d and Supplementary Table 5). By contrast, after only 20 K cycles, the ECSAs and MA for commercial Pt/C, Pt NWs/C, PtNi NWs/C, PtNiMo NWs/C, and PtNiAu NWs/C catalysts severely decrease 68.4% and 70.2%, 30.8% and 34.5%, 49.4% and 53.5%, 19.2% and 22.8%, 14.4% and 18.3% (Fig. 1e and Supplementary Fig. 21). The structure and composition of PtNi-MoAu NWs/C catalyst after 80 K of ADT were further probed. The

STEM image demonstrates the well-maintained morphology of PtNi-MoAu NWs/C after 80 K cycles (Fig. 5a), whereas the commercial Pt/C and other Pt-based NWs/C catalysts exhibit varying degrees of sintering after only 20 K cycles (Supplementary Fig. 8, 22a and 23). Its diameter only increases from 1.1 nm (initial) to 1.4 nm (after 80 K cycles) (Fig. 5a and Supplementary Fig. 22b), consistent with the ECSAs losses. And the EDS mapping and line-scan profiles further confirm its well-reserved elemental distribution and composition (Fig. 5b, c). The atomic ratio of Pt:Ni:Mo:Au (3.65:1.00:0.16:0.20) in PtNiMoAu NWs/C catalyst after 80 K cycles is relatively close to its initial composition (3.02:1.00:0.11:0.13), contrasting with the big change for PtNi NWs/C catalyst after only 20 K cycles from 3.03:1.00 to 11.01:1.00. The dissolution of Pt and Ni for PtNiMoAu NWs/C catalyst at different ADT cycles was further examined based on ex-situ ICP-MS measurements. Self-consistent with our expectation from the distinct roles of Mo and Au dopants, the PtNiMoAu NWs/C presents the lowest leaching mass of Pt and Ni metal among all PtNi-based NWs (Fig. 5d).

Besides, in-situ XANES was carried out to experimentally support the remarkable stability of the PtNiMoAu NWs/C catalyst for ORR. From XANES spectra of the Pt $L_3$ edge, we can find that the normalized white line intensity ($\mu_{norm}$) of PtNiMoAu NWs slightly increases with

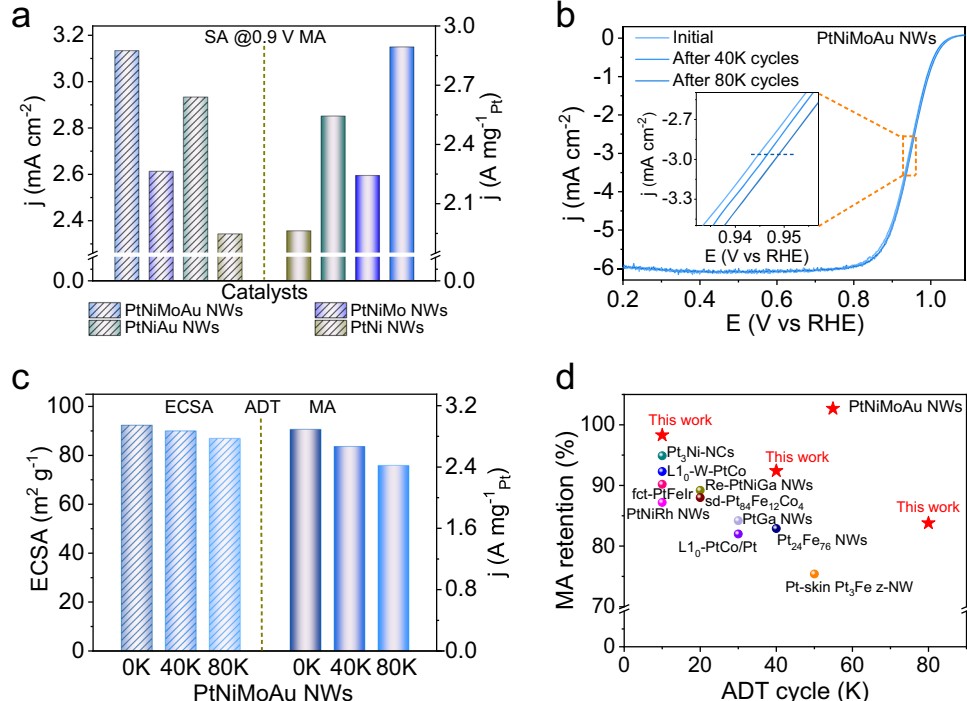

**Fig. 4 | Electrochemical properties of different catalysts. a** The SA and MA of different catalysts at 0.9 $V_{RHE}$. **b** ORR polarization curves of PtNiMoAu NWs/C catalyst before and after ADT for different cycles. The inset showed an enlarged view of the area marked by the orange square. **c** The comparison of ECSAs and MA for PtNiMoAu NWs/C catalyst at 0.9 $V_{RHE}$ before and after ADT for different cycles. **d** Comparison of MA retention for recently reported catalysts after different cycles of ADT.

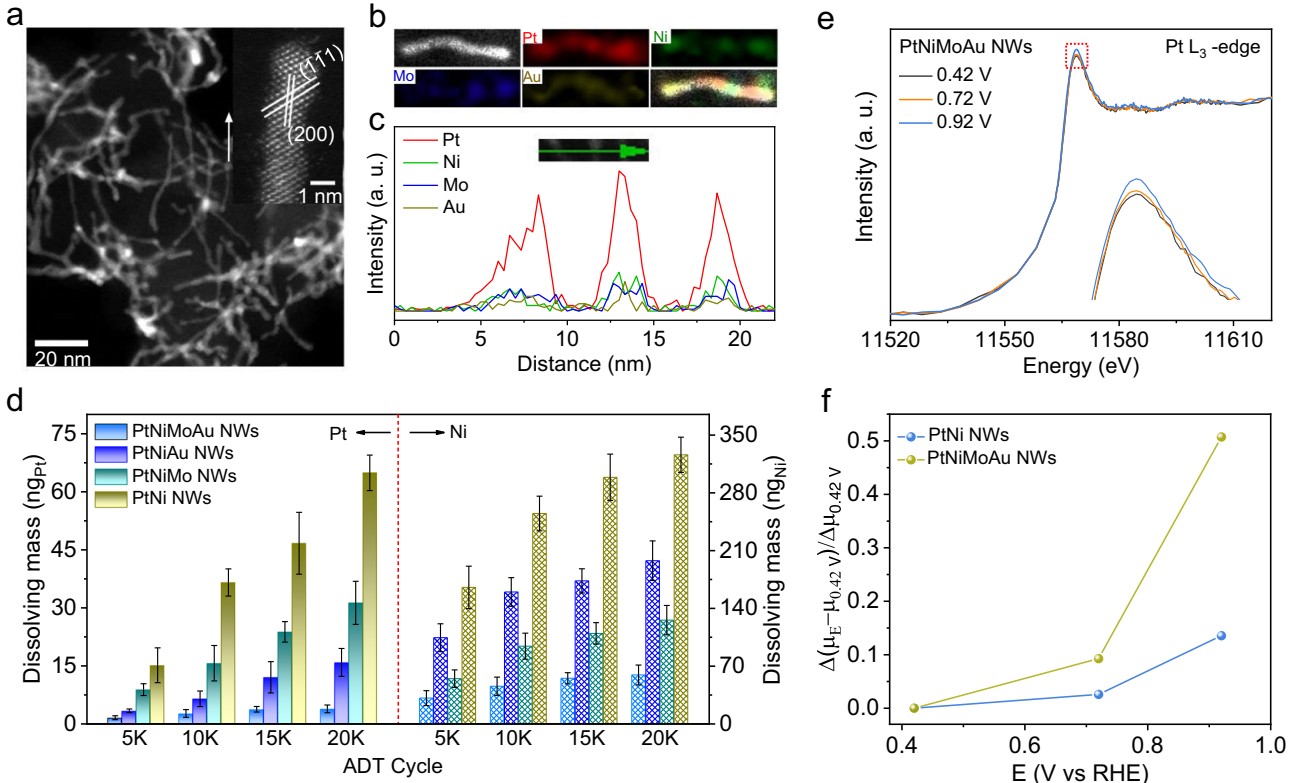

**Fig. 5 | Structural characterizations and in-situ XANES analysis. a** Low-magnification STEM image after 80 K of ADT. The inset showed atomically resolved HAADF-STEM image. **b** STEM-EDS elemental mapping profiles and **c** EDS line-scanning profiles after 80 K of ADT. **d** Dissolving mass of Pt/Ni for PtNi-based NWs after different cycles. The dissolving mass values are means with error bars (standard deviations) from three replicates. **e** In situ Pt $L_3$-edge XANES spectra of PtNiMoAu NWs at different potentials. The inset showed an enlarged view of the area marked by the red square. **f** Comparison of the change of the Pt adsorption edge peaks ($\Delta\mu$) of the XANES spectra (relative to $\Delta\mu$ at 0.42 $V_{RHE}$) for PtNi NWs and PtNiMoAu NWs as a function of potential obtained in 0.1 M $HClO_4$.

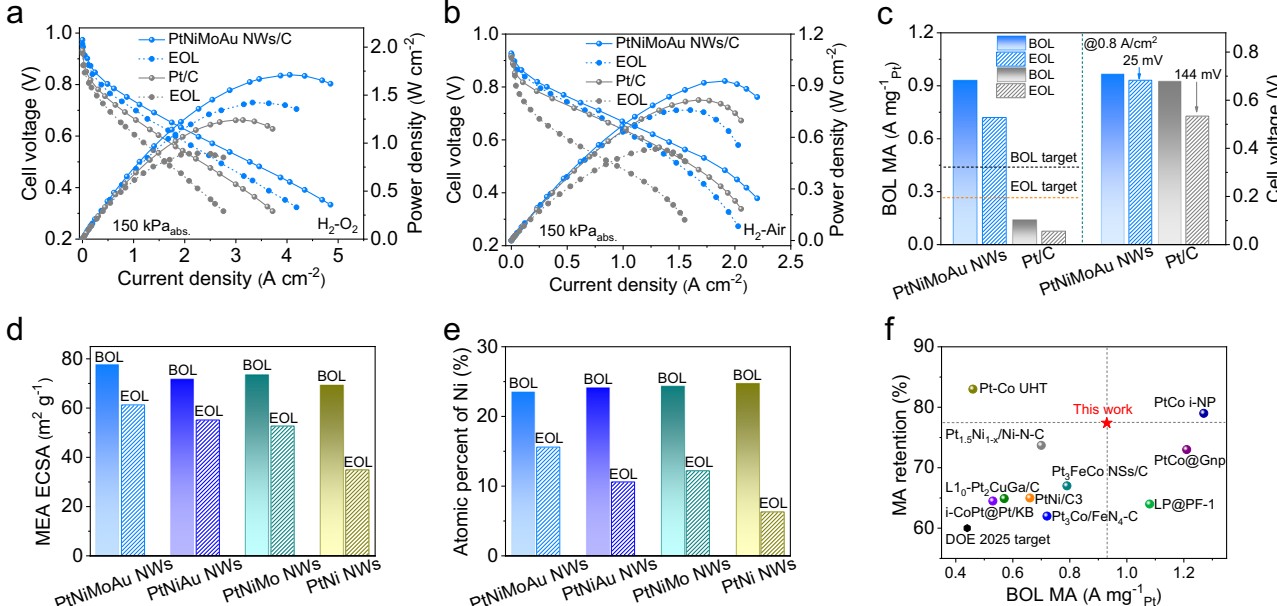

**Fig. 6 | Fuel cell performance of the PtNiMoAu NWs/C catalyst. a** $H_2$-$O_2$ fuel cell polarization curves and power density curves before and after 30 K cycles of ADT. Anode: $H_2$ flow rate = 200 mL min$^{-1}$, 0.05 mg$_{Pt}$ cm$^{-2}$ for Pt/C; Cathode: $O_2$ flow rate = 500 mL min$^{-1}$, 0.10 mg$_{Pt}$ cm$^{-2}$ for PtNiMoAu NWs/C and 0.12 mg$_{Pt}$ cm$^{-2}$ for Pt/C. **b** $H_2$-Air fuel cell polarization curves and power density curves of PtNiMoAu NWs/C and Pt/C before and after 30 K cycles of ADT. Anode: $H_2$ flow rate = 200 mL min$^{-1}$; Cathode: Air flow rate = 500 mL min$^{-1}$. **c** Changes of MA (left, $H_2$-$O_2$, black and orange dash lines indicate BOL and EOL of DOE targets, respectively) and cell voltage at 0.8 A cm$^{-1}$ (right, $H_2$-Air) of PtNiMoAu NWs/C and Pt/C before and after 30 K cycles of ADT. **d** The changes of MEA ECSAs and **e** atomic percent of Ni for PtNiMoAu NWs/C, PtNiAu NWs/C, PtNiMo NWs/C, and PtNi NWs/C before and after 30 K cycles of ADT. **f** Comparison of BOL MA and MA retention (after 30K-cycle ADT) with those typical catalysts reported recently.

the potential (Fig. 5e), while the PtNi NWs display a much larger increase at the same operational condition (Supplementary Fig. 24). The difference is more clear in the relative change of the white line intensity [$(\Delta\mu_E$-$\Delta\mu_{0.42VRHE})/\Delta\mu_{0.42VRHE})$] of the Pt L$_3$ edge spectra for PtNiMoAu NWs/C and PtNi NWs/C catalysts as a function of applied potential[20,39]. As shown in Fig. 5f, the increase in white line intensity for PtNiMoAu NWs/C catalyst occurs at a higher applied potential compared with that for PtNi NWs/C catalyst, corroborating the elevated Pt oxidation potential for PtNiMoAu NWs/C catalyst. In addition, anodic shift of Pt/Pt$^{2+}$ redox potential for PtNiMoAu NWs/C catalyst also supports this result (Supplementary Fig. 25). This finding is indeed consistent with the electron-rich Pt atoms for PtNiMoAu NWs as demonstrated by the XANES spectrum (Fig. 3h) and Bader analysis (Supplementary Table 3), which could curb the Pt dissolution and in turn stabilize the catalyst. All these preceding evidences strongly support our conceptual design that integrating the different functions of Mo and Au into PtNi NWs would maximize the catalyst stability.

## Fuel cell performance
We further fabricated the membrane electrode assembly (MEA) to evaluate the fuel cell performance of PtNi NWs/C, PtNiMo NWs/C, PtNiAu NWs/C, and PtNiMoAu NWs/C. The polarization curves and power density curves were recorded in $H_2$-$O_2$ (Fig. 6a) and $H_2$-Air environment (Fig. 6b) (80 °C and 150 kPa absolute pressure), in which the Pt loadings for Pt/C as anode catalyst and PtNi-based NWs/C as cathode catalyst were 0.05 mg$_{Pt}$ cm$^{-2}$ and 0.10 mg$_{Pt}$ cm$^{-2}$, respectively (Pt/C catalyst with Pt loading of 0.12 mg$_{Pt}$ cm$^{-2}$ on the cathode was also employed as the reference). Before evaluating MEA performance of catalysts, we first tested the pressure drop between the inlet and outlet at different flow rates (Supplementary Fig. 26), in which the gap pressure drop of anode ($H_2$, 200 mL min$^{-1}$) and cathode ($O_2$, 500 mL min$^{-1}$) is severally 3 kPa and 6 kPa. Given the relatively low pressure drop, we reasonably believe that it has a negligible impact on MEA performance. The $H_2$−$O_2$/$H_2$-Air fuel cell assembled with

PtNiMoAu NWs/C cathode shows a peak power density of 1.71/ 0.93 W cm$^{-2}$, exceeding the corresponding values of 1.24/0.82 W cm$^{-2}$ for commercial Pt/C. Besides, the PtNiMoAu NWs/C presents a 6.6-, 2.0-, 1.3-, and 1.7-fold enhancement in the beginning-of-life (BOL) MA (0.93 A mg$^{-1}_{Pt}$) relative to the Pt/C (0.14 A mg$^{-1}_{Pt}$), PtNi NWs/C (0.46 A mg$^{-1}_{Pt}$), PtNiMo NWs/C (0.70 A mg$^{-1}_{Pt}$), and PtNiAu NWs/C (0.55 A mg$^{-1}_{Pt}$) at 0.9 $V_{iR\text{-}free}$ (Fig. 6c and Supplementary Fig. 27). Even comparing with the value that was set in the 2025 targets by U.S. Department of Energy (DOE) (MA of 0.44 A mg$^{-1}_{Pt}$), the remarkable mass activity of PtNiMoAu NWs/C catalyst presents 2.1-fold increasement, showing the great potential for PEMFCs. Moreover, the PtNiMoAu NWs/C under the $H_2$-Air condition delivers the improved current density of 0.35 A cm$^{-2}$ at the kinetic dominant region of 0.8 V when compared to the Pt/C (0.16 A cm$^{-2}$ at 0.8 V) and the DOE 2025 target (0.30 A cm$^{-2}$) (Supplementary Fig. 28). For ADT, the standard square-wave protocol by holding the cathode at 0.60 V for 3 s and 0.95 V for 3 s was employed to evaluate the durability of catalysts (see Methods). The end-of-life (EOL) MA at 0.9 $V_{iR\text{-}free}$ ($H_2$-$O_2$) and a voltage loss at 0.8 A cm$^{-2}$ ($H_2$-Air) for PtNiMoAu NWs/C are 0.72 A mg$^{-1}_{Pt}$ (22.6% MA loss) and 25 mV after 30 K cycles of ADT, respectively, which comprehensively exceed the DOE's 2025 durability goal (MA loss <40% and voltage loss <30 mV). As a contrast, the Pt/C, PtNi NWs/C, PtNiMo NWs/C, and PtNiAu NWs/C show substantial MA loss of 45.7%, 58.7%, 41.4%, and 32.7%, respectively. Besides, the variation trend of ECSAs, voltages loss ($H_2$-$O_2$), Ni content, and morphology of PtNi NWs/C, PtNiAu NWs/C, PtNiMo NWs/C, and PtNiMoAu NWs/C catalysts during the MEA durability tests also was in accordance with the results obtained under the rotating disk electrode measurements (Fig. 6d, e, Supplementary Figs. 29−31, and Supplementary Table 6). Even comparing with the Pt-based catalysts reported recently, the excellent BOL MA and durability of PtNiMoAu NWs/C indeed render it as the top catalyst for durable fuel cells (Fig. 6f and Supplementary Table 7). Beyond this, we can envision our delicate design would also enhance the catalyst durability a more challenging start-up and shutdown

process of the fuel cells (generally under the applied potential over 1.0 V) through mitigation of the sintering process. All these preceding results together strongly evidence the successful design of active and durable PtNiMoAu NWs/C catalyst for practical fuel cells.

## Discussion

To sum up, we have identified the distinct roles of Mo and Au dopants in inhibiting the dissolution of Ni and Pt metal, respectively, in terms of PtNi-based NWs/C as model catalysts, by combining the experimental evidences and DFT calculations. Insightful studies indicated that the distinct roles of Mo and Au dopants are essentially derived from the stronger interaction of atomic orbital for Pt-Au (*d-d*) and Ni-Mo (*d-d*). Based on this atomistic understanding, we delicately designed a remarkable PtNiMoAu NWs/C catalyst that possesses a concurrently high MA of 2.89 A mg$^{-1}_{Pt}$ and preeminent stability, with only 16.2% loss of MA after 80 K cycles of ADT. By contrast, PtNi NWs/C catalyst showed a 53.5% loss of MA after only 20 K cycles of ADT, persuasively confirming the advantage of Mo/Au co-doping in improving its ORR stability. Moreover, when assembling the PtNiMoAu NWs/C catalyst into the fuel cell cathode, a high MA retention of 77.4% (H$_2$-O$_2$, 0.9 V$_{iR-free}$) and a low voltage loss of 25 mV (H$_2$-Air, 0.8 A cm$^{-2}$) after ADT were output, proving the highly durable fuel cell performance. This work advances the design of robust PtM catalysts to a more precise stage by providing insights into the functions of dopants, which is thus of general importance for the fields of catalysis, sensing, and even beyond.

## Methods

### Chemicals

Platinum (II) acetylacetonate (Pt(acac)$_2$, 97%), nickel (II) acetylacetonate (Ni(acac)$_2$, 97%), molybdenum (III) acetylacetonate (Mo(acac)$_3$, 97%), gold (III) chloride trihydrate (HAuCl$_4$·3H$_2$O, 99%), cetyltrimethylammonium bromide (CTAB, 99%), tungsten hexacarbonyl (W(CO)$_6$, 97%), oleylamine (OAm, 70%) and Nafion (5 wt%) were purchased from Sigma-Aldrich. Ethanol (CH$_3$CH$_2$OH, 99%) and perchloric acid (HClO$_4$, 70–72%) were purchased from Sinopharm Chemical Reagent Co. Ltd. (Shanghai, China). The water (18.2 MΩ/cm) was freshly prepared through an ultra-pure purification system (Master-515Q, HHitech). All the chemicals were used without further purification.

### Synthesis of ultrathin Pt-based nanowires

In a typical synthesis of ultrathin PtNiMoAu NWs, Pt(acac)$_2$ (20 mg), Ni(acac)$_2$ (10 mg), Mo(acac)$_3$ (8 mg), HAuCl$_4$·4H$_2$O (2.7 mg), CTAB (75 mg), and 4 mL OAm were added into a 30 mL glass vial. After sonicated for 30 min, W(CO)$_6$ (20 mg) was added into the pre-dispersed solution and capped. The resulting homogeneous mixture was then heated and kept at 175 °C for 2 h in an oil bath. Finally, the products were collected by centrifugation at 13000 × g and cleaned four times with a hexane/ethanol mixture (v/v = 2/1), then dried under vacuum. The preparation of other ultrathin Pt-based nanostructures was similar to that of PtNiMoAu NWs except that Mo(acac)$_3$ or HAuCl$_4$·4H$_2$O were taken out and the reaction temperature was changed. The detailed synthetic parameters for other Pt-based NWs have been listed in Supplementary Table 1.

### Characterization techniques

XRD patterns were collected to analyze the crystal structures of NWs by X-ray diffractometer (Rigaku Miniflex-600) with Cu K$\alpha$ radiation ($\lambda$ = 0.15406 nm, 40 kV). TEM images were carried out on a JEOL 2100-Plus operating at 120 kV with the samples deposited on carbon-coated copper TEM grids. HAADF-STEM images and EDS line-scan file were taken on a JEOL ARM-200F microscope with a spherical aberration corrector operating at 200 kV. Elemental analysis of ultrathin NWs was quantitatively determined by ICP-MS with a SPECTRO BLUE SOP. The XPS spectra were collected using an Escalab 250Xi equipped with an Al

Ka (1486.6 eV) excitation source. All the spectra collected were corrected using a Shirley background. All XAFS data were collected at BL14B station of Shanghai Synchrotron Radiation Facility (SSRF), China. The storage ring of SSRF operates at 3.5 GeV with a maximum current of 210 mA. The electrochemical in situ XAFS measurements were carried out in a custom-fabricated three-electrode system with a 1.4 × 0.7 cm$^2$ carbon cloth loaded with PtNi and PtNiMoAu NWs catalysts as the working electrode and 0.1 M continuously O$_2$-saturated HClO$_4$ solution as the electrolyte. For the in situ XAFS spectral data acquisition of Pt L$_3$-edge (11564 eV), we calibrated the positions of absorption edges (E$_0$) by using Pt foil and Fe foil standard samples, respectively, and all spectra were collected in the same beam time by fluorescence mode to ensure comparability.

### Preparation of working electrode

For different NWs/C catalysts, 4 mg of the prepared NWs was added to a chloroform solution (8 mL) and sonicated for 1 h. The above-dispersed solution was dropwise added to an ethanol solution containing 16 mg of carbon support (Vulcan XC-72) under vigorous stirring for 30 min. After centrifugation and washing twice with hexanes by centrifugation, the NWs/C catalysts were re-dispersed in acetic acid and then heated at 70 °C for 12 h to remove the surfactants on the surface of NWs. A certain amount of as-obtained catalyst was mixed with 0.5 mL isopropanol, 0.495 mL ethanol, and 0.005 mL Nafion (5 wt %) and sonicated for 1 h to form the homogenously mixed catalyst ink solution. For the commercial Pt/C catalysts (20 wt% loading on Vulcan XC-72 carbon support, Johnson Matthey), the ink solution (2 mg/mL) was prepared and sonicated for 1 h. Finally, prepared catalyst ink was dropped onto the glassy carbon rotating disk electrode (GC, RDE with geometric area of 0.196 cm$^2$) with the loading amount of Pt at 2.0 μg, 2.1 μg, 2.2 μg, 2.0 μg, 2.3 μg, and 4.0 μg for PtNiMoAu NWs/C, PtNiAu NWs/C, PtNiMo NWs/C, PtNi NWs/C, Pt NWs/C, and commercial Pt/C catalysts, respectively (based on ICP-MS).

### Electrochemical testing

Electrochemical tests were conducted using a three-electrode cell on a CHI760e electrochemical workstation (Chenhua Instrument, China). A glassy carbon Rotating Disk Electrode (RDE, diameter: 5 mm) was used as the working electrode, the Ag/AgCl (3 M KCl) electrode was used as a reference electrode, and a platinum wire was used as a counter electrode. All potentials were converted to the reversible hydrogen electrode reference. The CVs were tested in a N$_2$-saturated 0.1 M HClO$_4$ electrolyte with a sweep rate of 50 mV s$^{-1}$. The ECSAs were calculated by integrating the hydrogen adsorption/desorption charge area between 0.05 and 0.38 V$_{RHE}$ from the CVs. The equation for the calculation of ECSAs is shown as follows:

$$ECSAs(m^2/g_{Pt}) = \left[ \frac{Q_{H-adsorption}(C)}{210\mu C/cm^2 L_{Pt}(mg_{Pt}/cm^2)A_g(cm^2)} \right] \times 10^5 \quad (1)$$

The charge density for the adsorption of one monolayer hydrogen on Pt (Q$_H$) was assumed to be 210 μC cm$^{-2}$. L$_{Pt}$ (mg$_{Pt}$ cm$^{-2}$) was the working electrode Pt loading. A$_g$ (cm$^2$) was the geometric surface area of the glassy carbon electrode (0.196 cm$^2$). The ORR polarization curves were measured in an O$_2$-saturated 0.1 M HClO$_4$ solution between 0.05 and 1.05 V$_{RHE}$ using a sweep rate of 10 mV s$^{-1}$ at a rotation rate of 1600 rpm. The ADTs were performed via cyclic sweeps between 0.6 and 1.0 V$_{RHE}$ in an O2-saturated 0.1 M HClO$_4$ electrolyte at a sweep rate of 100 mV s$^{-1}$ for different cycles.

### MEA fabrication and test

The catalytic activity was also evaluated under PEMFC operating conditions. Specifically, each catalyst was mixed with isopropanol, deionized water, and Nafion solution by ultrasonicating for 1 h to form homogeneous ink with the ratio of ionomer to carbon (I/C) of 0.8. The

ink was then sprayed the ink onto the Nafion® 211 membrane (DuPont). The catalyst-coated-membrane (CCM) with an active geometric area of 12.25 cm$^2$ was applied to the gas diffusion layer (GDL, Toray TGP-H-060). The compression ratio of GDL was calculated to be 31.5%. PtNi-MoAu NWs/C, PtNiAu NWs/C, PtNiMo NWs/C, PtNi NWs/C, and commercial Pt/C were employed as the cathode catalysts and Pt/C (20 wt% loading, Johnson Matthey) was used for the anode. The Pt loading at the cathode was 0.10 mg$_{Pt}$ cm$^{-2}$ for PtNi-based NWs/C and 0.12 mg$_{Pt}$ cm$^{-2}$ for commercial Pt/C, respectively. As for the anode, Pt/C was used with a loading of 0.05 mg$_{Pt}$ cm$^{-2}$. Fuel cell testing was performed in a single cell using a commercial fuel cell test system (Scribner 850e, Hephas Energy Corporation). The MEA was sandwiched between two graphite plates with single serpentine flow channels. The corresponding pressure drop across the cathode and anode was measured using two pressure sensors connected to the inlet and outlet of the cell, respectively, while the cell was operated at 80 °C, 150 kPa (absolute, abs) back pressure, relative humidity (RH) 100%, H$_2$ flow rate of 200 mL min$^{-1}$ and O$_2$/Air flow rate of 500 mL min$^{-1}$. Fuel cell polarization curves were recorded using potential step mode with 50 mV/point (holding 2 min for each point). The MA was calculated by normalizing the measuring currents with Pt loading amount in 150 kPa$_{abs}$ H$_2$/O$_2$ (80 °C, 100% RH, 200$_{anode}$/500$_{cathode}$ sccm) at 0.9 V$_{iR-free}$. The ADTs were conducted by the standard 30 K-cycle square-wave protocol to evaluate the durability of catalysts. Specifically, the cathode was held at 0.60 V for 3 s and 0.95 V for 3 s in each cycle.

## DFT calculations

All of the DFT calculations were performed using the Vienna Ab-initio simulation package program[40–42], which uses a plane-wave basis set and a projector augmented wave method (PAW) for the treatment of core electrons[41]. The Perdew, Burke, and Ernzerhof exchange-correlation functional within a generalized gradient approximation (GGA-PBE)[43] was used in our calculations, and the van der Waals (vdW) correction proposed by Grimme (DFT-D3)[44] was employed due to its good description of long-range vdW interactions. For the expansion of wavefunctions over the plane-wave basis set, a converged cutoff was set to 450 eV.

In order to simulate the Pt NWs and PtNi bimetallic NWs, seven-layer 2×2 Pt (111) (a = b = 5.456 Å) and 2 × 2 Pt$_3$Ni (111) (a = b = 5.383 Å) slabs with periodical boundary conditions were used, respectively. Besides, to model the PtNiMo trimetallic NWs and PtNiMoAu tetra-metallic NWs, a Pt atom in the subsurface layers of the Pt$_3$Ni (111) slab was substituted by a Mo atom and two Pt atoms in the subsurface layers of the Pt$_3$Ni (111) slab were substituted by a Mo and an Au atom. Moreover, considering the surface Ni and Mo atoms will be leached into an acidic solution, the outmost layers of the Pt$_3$Ni (111), Mo-doped Pt$_3$Ni (111), and Mo- and Au-doped Pt$_3$Ni (111) slabs were all replaced as shown in Supplementary Fig. 10. The vacuum space was set to 17 Å in the z direction to avoid interactions between periodic images. In geometry optimizations, all the atomic coordinates were fully relaxed up to the residual atomic forces smaller than 1 × 10$^{-4}$ eV/Å, and the total energy was converged to 10$^{-5}$ eV. The Brillouin zone integration was performed on the (9 × 9 × 1) Monkhorst–Pack k-point mesh[45].

The ORR pathways on various NWs were calculated in detail according to the electrochemical framework developed by Nørskov and his co-workers[46,47]. For ORR, the four-electron reaction mechanism follows several elementary steps:

$$O_2(g) + * \rightarrow O_2^* \tag{2}$$

$$O_2^* + H^+ + e^- \rightarrow OOH^* \tag{3}$$

$$OOH^* + H^+ + e^- \rightarrow O^* + H_2O(l) \tag{4}$$

$$O^* + H^+ + e^- \rightarrow OH^* \tag{5}$$

$$OH^* + H^+ + e^- \rightarrow H_2O(l) + * \tag{6}$$

where the * represents the active site on the electrocatalyst surface, (l) and (g) refer to liquid and gas phases, respectively, and OOH$^*$, O$^*$ and OH$^*$ are adsorbed intermediates.

The binding energies of OOH$^*$, O$^*$ and OH$^*$ were obtained by DFT calculations as follows:[46,47]

$$\Delta E_{OOH^*} = E(OOH^*) - E(*) - \left(2E_{H_2O} - 3/2E_{H_2}\right) \tag{7}$$

$$\Delta E_{O^*} = E(O^*) - E(*) - \left(E_{H_2O} - E_{H_2}\right) \tag{8}$$

$$\Delta E_{OH^*} = E(OH^*) - E(*) - \left(E_{H_2O} - 1/2E_{H_2}\right) \tag{9}$$

in which, E(*), E(OOH$^*$), E(O$^*$), and E(OH$^*$) are the ground state energies of a clean surface and surfaces adsorbed with OOH$^*$, O$^*$ and OH$^*$, respectively. E$_{H_2O}$ and E$_{H_2}$ are the calculated DFT energies of H$_2$O and H$_2$ molecules in the gas phase. If we consider the zero point energy (ZPE) and entropy correction, the free energies of adsorption, $\Delta G_{ads}$, can be transformed from DFT binding energies, $\Delta E_{ads}$, as follows:

$$\Delta G_{ads} = \Delta E_{ads} + \Delta ZPE - T\Delta S + eU \tag{10}$$

where $\Delta E_{ads}$ is the binding energy of adsorption species OOH$^*$, O$^*$ and OH$^*$. $\Delta$ZPE, $\Delta$S, U and e are the ZPE changes, entropy changes, applied potential at the electrode, and charge transferred.

Using the adsorption free energies obtained from (10) and (7)-(9), the reaction free energies of ORR reactions (2)-(6) can be calculated as:

$$\Delta G_1 = \Delta G_{OOH^*} - 4.92 \tag{11}$$

$$\Delta G_2 = \Delta G_{O^*} - \Delta G_{OOH^*} \tag{12}$$

$$\Delta G_3 = \Delta G_{OH^*} - \Delta G_{O^*} \tag{13}$$

$$\Delta G_4 = -\Delta G_{OH^*} \tag{14}$$

Thus, for the ORR reactions, the onset potential, U$_{RHE}^{onset}$, and the overpotential, $\eta^{ORR}$, can be expressed as:[46,47]

$$U_{RHE}^{onset} = -\max\{\Delta G_1, \Delta G_2, \Delta G_3, \Delta G_4\} \tag{15}$$

$$\eta^{ORR} = 1.23V - U_{RHE}^{onset}/e \tag{16}$$

The vacancy formation energy is calculated as:

$$E_{V_{Pt}} = E_{slab-V_{Pt}} + \mu_{Pt} - E_{slab} \tag{17}$$

where E$_{slab-VPt}$ represents the energy of a slab with one Pt vacancy, $\mu_{Pt}$ represents the chemical energy of a Pt atom, E$_{slab}$ represents the energy of a perfect slab.

We also calculated the vacancy formation energy of Ni (E$_{slab-VNi}$) in PtNi slab, PtNiMo slab, PtNiAu slab and PtNiMoAu slab by formula (17), where E$_{slab-VNi}$ represents the energy of a slab with one Ni vacancy on the subsurface layer, $\mu_{Ni}$ represents the chemical energy of a Ni atom, E$_{slab}$ represents the energy of a perfect slab. In addition, we constructed 2 × 2 Pt$_3$Ni conventional cells with a Ni vacancy model to

calculate the Ni diffusion energy. The bulk Ni diffusion was proposed to diffuse to the nearest Ni vacancy site and the corresponding transition state searches were conducted with the climbing image nudged elastic band method (CI-NEB) method. For the Ni diffusion energy in the PtNiMo slab and PtNiAu slab, a Pt atom near the Ni diffusion pathway of the $Pt_3Ni$ (111) slab was substituted by the Mo atom and Au atom, respectively. For the Ni diffusion energy in the PtNiMoAu slab, two Pt atoms near the Ni diffusion pathway of the $Pt_3Ni$ (111) slab were substituted by the Mo atom and Au atom.

## Data availability
The data that support the plots within this paper and other finding of this study are available from the corresponding authors upon request. Source data are provided with this paper.

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

## Acknowledgements

This work was supported by the National Key Research and Development Program of China (No. 2021YFA1502000 to H.H.), NSFC (Nos. 22322902, U22A20396, 22211540385, and 22309050 to H.H. and L.G.), the Science and Technology Innovation Program of Hunan Province (No. 2021RC3065, 2021RC2053, and 2023JJ40117 to H.H. and L.G.), the Jiebang Guashuai Project of Changsha City (Grant No. kq2301009 to H.H.), the China Postdoctoral Science Foundation (2023T160205 and 2023M741118 to L.G.), Shenzhen Science and Technology Program (Nos. JCYJ20210324120800002 and JCYJ20220818100012025 to H.H.). The X-ray ab-sorption structure (XAS) spectra were performed at the BL11B beam line of the Shanghai Synchrotron Radiation Facility (SSRF).

## Author contributions

L.G. and H.H. conceived the idea and designed experiments. L.G. performed the preparation of samples, carried out the electrochemical measurements, and analyzed the experimental data. Z.Y. and W.Z. conducted the DFT simulation and theoretical analyses. T.S. and M.L. conducted the TEM and EDX characterizations. X.C., W.L. and Q.Y. helped with the analysis and discussion of experimental data. H.H. wrote the manuscript. All the authors were involved in the discussion and analysis of the manuscript.

## Competing interests

The authors declare no competing interests.
