## [Peer Review File · Nature Communications]

Identifying the distinct roles of dual dopants in stabilizing the platinum-nickel nanowire catalyst for durable fuel cellREVIEWER COMMENTS

Reviewer #1 (Remarks to the Author):

Gao et al. reported the synthesis of Mo and Au codoped PtNi nanowires and their performance for cathodic oxygen reduction catalysts in PEMFCs. It has been widely reported that the third metal doping (typically, Au, Mo, Rh, et al.) could improve the stability of Pt/C and PtNi/PtCo alloy catalysts. Although the authors got some new understandings on how Mo and Au dopants work in improving the stability, I do not think this work is important enough for publication in Nature Communications. One of the key conclusions of this work is that the co-doping of Mo and Au can greatly improve the durability of PtNi catalysts in a synergetic manner. But the authors did not prove this issue in MEA, which is more relevant to practical PEMFCs applications than RDE. The durability test of the PtNi, Au-PtNi, Mo-PtNi, and AuMo-PtNi catalysts in MEA is necessary to confirm the co-doping effect. The changes of ECSA and Ni content of these catalysts during the MEA durability tests should be demonstrated. The related MA, oxygen/proton transport resistance and voltage loss should be analyzed.

The authors claimed that "the MA of the PtNiMoAu NWs/C catalyst presents even 6.6-fold increase compared to the value that was set in the 2025 targets by U.S. Department of Energy (DOE)". Such conclusion is misleading, as the MA target of US DOE refers to the MEA test instead of RDE.

Reviewer #2 (Remarks to the Author):

The authors report a study of dual doping PtNi alloy catalysts with combination of Mo and Au to understand their role in enhancing durability in a fuel cell. While the use of these dopants/alloying elements is not new, the detailed study is quite novel and the results will have a significant impact in the development of electrocatalysts for fuel cells and electrolyzers. Overall the paper is well written, and the experimental work seems to be very well done. The performance of their PtNiMoAu/C catalysts are quite impressive. The paper needs some modest improvement in the presentation of the data and discussion of related literature as detailed below.

Detailed comments

1. Figure 1e is a very important figure in this work, but it is extremely difficult to read (even in full color). I'd recommend the changes in MA and ECSA be on separate plots. I'd also recommend adding a plot of the retained MA/ECSA for each catalyst (perhaps add to SI as part of this data is in Table S4). This will make it easier for the reader to compare with other published literature.
2. Likewise, a table that summarizes the beginning of life values for MA & ECSA from RDE would be extremely useful for the readers.
3. The introduction nicely covers relevant literature alloying Pt with Mo as an electrocatalyst. However, the authors should also discuss recent literature from Esfahani where Mo or Mo + Si has been doped into metal oxide support materials and found to also greatly enhance Pt-based fuel cell electrocatalyst stability (doi:10.1016/j.apcatb.2016.08.041, doi:10.1016/j.apcatb.2019.118272, doi:10.1016/j.apcatb.2020.118743). In particular, they noted the suppression of Pt particle size growth is greatly suppressed in these doped metal oxides through what is often referred to as "strong metal support interaction" (SMSI), which can also inhibit the sintering and leaching of active metals. The SMSI is often a general term to account for Fermi level interactions between metal oxide support and catalytic nanoparticles. While the SMSI is common for many different metal oxide supports for fuel cells, but the durability reported in 10.1016/j.apcatb.2020.118743 for a dual doped support that contained Mo + another dopant (Si) is also impressive. Thus, this reviewer wonders if mechanism of stabilization for Mo-doped electrocatalysts reported by the authors could be similar for Mo-doped supports. A short discussion of how these results could also benefit doped support design could make this paper have a greater impact in the field.

4. Page 11, line 298: Please add “, respectively” after 0.05 mgPt cm⁻² and 0.10 mgPt cm⁻²
5. Durability testing: The authors have tested all of their catalysts using variations of the DoE “load cycle” protocols, with the highest UPL of 1.0V vs RHE. Have the authors studied the durability of these catalysts using higher UPLS like those in the DoE “start-up/shutdown” test (UPL of 1.5V vs RHE)? There is already a massive amount of data in this paper, so I do not expect the authors to perform new experiments. However, these higher potentials do accelerate the rate of Pt NP growth and agglomeration compared to the load cycle protocol (though the data is often convoluted by the high rates of carbon corrosion). Thus, it would also be useful if the authors discussed/hypothesized about the impact of performing this work at higher UPLs, particularly since the development of more corrosion stable carbon and non-carbon supports is a highly active area of study.
6. Durability Testing: On page 14: the authors state that the upper potential limit (UPL) used in their durability tests is 1.0 V vs RHE. However, Table S4 states the UPL is 1.1 V vs RHE. One of these is incorrect (most likely the Table) and needs to be fixed.

Reviewer #3 (Remarks to the Author):

The manuscript by Gao et al. reports on the study of PtNiAuMo alloys as the catalysts for the oxygen reduction for the application in fuel cells and clarifies the role of Au and Mo dopants on the enhancement of the activity and stability of the multimetallic alloy. The system described in the work demonstrates the performance surpassing that of the benchmark Pt/C and PtNi catalysts. The manuscript is well-structured and can be recommended for publication in Nature Communications once the following concerns are addressed:

- 1) The results of DFT calculations for ORR on Pt(111) discussed on P. 7-8 and Fig. S13 are significantly different from the published data on Pt(111) slab: in a seminal study [Phys. Chem. Chem. Phys., 2008, 10, 3722], the overpotential for ORR on Pt(111) is 0.75 eV which is significantly lower than the values reported in this manuscript not only for Pt, but also for PtNi, PtNiMo, PtNiAu. Referee acknowledges that different calculation parameters will have an impact on the absolute energy values, yet such an inconsistency for the relatively simple and well-studied system like metallic platinum has to be carefully addressed to support the validity of the DFT data in the manuscript.
- 2) What is the reason for a decrease of the Pt²⁺ fraction in PtNiMoAu alloy as compared to PtNi (Fig. 3g)? Is it because a smaller fraction of Pt sites is located on the surface of PtNiMoAu? This is also likely the dominant reason for the lower intensity of the white line in Pt L-edge XAS spectrum, rather than the electronic effects from the neighboring atoms as discussed on P. 9.
- 3) Analysis of the in situ XANES data indicating that oxidation of Pt in PtNiMoAu occurs at higher potential than for PtNi should be supplemented with the analysis of CV data, which should demonstrate the anodic shift of Pt/Pt²⁺ redox potential. These changes of the formal redox potentials of platinum should be explained in terms of the inductive effect from Mo/Au.
- 4) Fig. S16 is absolutely confusing and should be removed as it represents a fitting of the noise.

Point-by-point response to reviewer comments

Manuscript ID: NCOMMS-23-28934

Title: Identifying the distinct roles of dual dopants in stabilizing the PtNi alloy catalyst for durable fuel cell

Reviewer #1

Gao et al. reported the synthesis of Mo and Au codoped PtNi nanowires and their performance for cathodic oxygen reduction catalysts in PEMFCs. It has been widely reported that the third metal doping (typically, Au, Mo, Rh, et al.) could improve the stability of Pt/C and PtNi/PtCo alloy catalysts. Although the authors got some new understandings on how Mo and Au dopants work in improving the stability, I do not think this work is important enough for publish in Nature Communications.

One of the key conclusions of this work is that the co-doping of Mo and Au can greatly improve the durability of PtNi catalysts in a synergetic manner. But the authors did not prove this issue in MEA, which is more relevant to practical PEMFCs applications than RDE. The durability test of the PtNi, Au-PtNi, Mo-PtNi, and AuMo-PtNi catalysts in MEA is necessary to confirm the co-doping effect. The changes of ECSA and Ni content of these catalysts during the MEA durability tests should be demonstrated. The related MA, oxygen/proton transport resistance and voltage loss should be analyzed.

The authors claimed that “the MA of the PtNiMoAu NWs/C catalyst presents even 6.6-fold increasement compared to the value that was set in the 2025 targets by U.S. Department of Energy (DOE)”. Such conclusion is misleading, as the MA target of US DOE refers to the MEA test instead of RDE.

Author reply: We genuinely thank this reviewer for his/her constructive comments, which greatly improve the manuscript. Specifically, we have made the following changes according to the comments.

- (1) We have complemented the MEA studies for all reference catalysts and systematically analyzed the catalyst durability under MEA measurements (Supplementary Figs. 26, 28 and 29; Supplementary Table 6). Consistently, both the MA and durability of the catalysts obtained under MEA measurements show the same trend with those under RDE tests. About the MA, the PtNiMoAu NWs/C presents a 6.6-, 2.0-, 1.3-, and 1.7-fold enhancement in the beginning-of-life (BOL) MA ($0.93 \text{ A mg}^{-1}\text{Pt}$) relative to the Pt/C ($0.14 \text{ A mg}^{-1}\text{Pt}$), PtNi NWs/C ($0.46 \text{ A mg}^{-1}\text{Pt}$), PtNiMo NWs/C ($0.70 \text{ A mg}^{-1}\text{Pt}$), and PtNiAu NWs/C ($0.55 \text{ A mg}^{-1}\text{Pt}$) at $0.9 \text{ V}_{\text{iR-free}}$ (Fig. 6c and Supplementary Fig. 26). Besides, the PtNiMoAu NWs/C also shows the highest MEA durability, presenting both lowest loss in MA and voltage ($@0.8 \text{ A cm}^{-2}$) when compared to other catalysts (Supplementary Figs. 26 and 29). The changes of ECSA and composition for PtNi, PtNiAu, PtNiMo, and PtNiMoAu catalysts during the MEA durability tests were also monitored, which could well rationalize the observed durability trend (Supplementary Figs. 28 and 29; Supplementary Table 6). Because this work just focuses on the Pt-based catalyst, we did not evaluate the oxygen/proton transport resistance under MEA measurements, which is generally related with the thickness of the catalyst layer, ionomer content, et al. ([10.1016/j.electacta.2019.135474](https://doi.org/10.1016/j.electacta.2019.135474); [10.1039/d1cy00882j](https://doi.org/10.1039/d1cy00882j)). We believe that these self-consistent results could strongly support our main conclusion of this manuscript. The relevant content has been added to our revised manuscript (**page 11, lines 9-11, 17-20, and 31; page 12, lines 1-5**).
- (2) We agree with the reviewer's viewpoint that the MA target of US DOE refers to the MEA test instead of RDE. As thus, we have corrected the relevant description in the revised manuscript (**page 11, lines 20-23**).

We hope the revised manuscript guided by these valuable comments would reassure the reviewer for the publication.

Reviewer #2

The authors report a study of dual doping PtNi alloy catalysts with combination of Mo and Au to understand their role in enhancing durability in a fuel cell. While the use of these dopants/alloying elements is not new, the detailed study is quite novel and the results will have a significant impact in the development of electrocatalysts for fuel cells and electrolyzers. Overall the paper is well

written, and the experimental work seems to be very well done. The performance of their PtNiMoAu/C catalysts are quite impressive. The paper needs some modest improvement in the presentation of the data and discussion of related literature as detailed below.

Author reply: We genuinely thank this reviewer for his/her constructive and positive comments.

1. Figure 1e is a very important figure in this work, but it is extremely difficult to read (even in full color). I'd recommend the changes in MA and ECSA be on separate plots. I'd also recommend adding a plot the retained MA/ECSA for each catalyst (perhaps add to SI as part of this data is in Table S4). This will make it easier for the reader to compare with other published literature.

Author reply: We genuinely thank this reviewer for his/her careful reading of our manuscript. As suggested, we have plotted the graphs for the MA and ECSA, as well as the retained MA/ECSA for each catalyst. The relevant figures/content have been revised/added in our revised manuscript (Fig. 1e,f; Supplementary Fig. 7; page 5, lines 25 and 26).

2. Likewise, a table that summarizes the beginning of life values for MA & ECSA from RDE would be extremely useful for the readers.

Author reply: As suggested, the beginning of life values for MA & ECSA from RDE were summarized in the SI (Supplementary Table 2).

3. The introduction nicely covers relevant literature alloying Pt with Mo as an electrocatalyst. However, the authors should also discuss recent literature from Esfahani where Mo or Mo + Si has been doped into metal oxide support materials and found to also greatly enhance Pt-based fuel cell electrocatalyst stability (doi:10.1016/j.apcatb.2016.08.041, doi:10.1016/j.apcatb.2019.118272, doi:10.1016/j.apcatb.2020.118743). In particular, they noted the suppression of Pt particle size growth is greatly suppressed in these doped metal oxides through what is often referred to as "strong metal support interaction" (SMSI), which can also inhibit the sintering and leaching of active metals. The SMSI is often a general term to account for fermi level interactions between metal oxide support and catalytic nanoparticles. While the SMSI is common for many different metal oxide supports for fuel cells, but the durability reported in 10.1016/j.apcatb.2020.118743 for a dual doped support that contained Mo + another dopant (Si)

is also impressive. Thus, this reviewer wonders if mechanism of stabilization for Mo-doped electrocatalysts reported by the authors could be similar for Mo-doped supports. A short discussion of how these results could also benefit doped support design could make this paper have a greater impact in the field.

Author reply: We genuinely thank this reviewer for his/her insightful comments. Agreeing with the reviewer's hypothesis, we also believe that the mechanism of Mo and Au dopants on stabilization of PtNi electrocatalysts is similar with Mo-doped supports. In essence, the stabilization effect can be attributed to the strong interaction between components in the catalysts. We have added the relevant discussions in the revised manuscript (**page 3, line 19; Refs. 23-25**).

4. Page 11, line 298: Please add “ , respectively” after $0.05 \text{ mg}_{\text{Pt}} \text{ cm}^{-2}$ and $0.10 \text{ mg}_{\text{Pt}} \text{ cm}^{-2}$.

Author reply: We have added “ , respectively” after $0.05 \text{ mg}_{\text{Pt}} \text{ cm}^{-2}$ and $0.10 \text{ mg}_{\text{Pt}} \text{ cm}^{-2}$. (**page 11, line 14**).

5. Durability testing: The authors have tested all of their catalysts using variations of the DoE “load cycle” protocols, with the highest UPL of 1.0V vs RHE. Have the authors studied the durability of these catalysts using higher UPLS like those in the DoE “start-up/shutdown” test (UPL of 1.5V vs RHE)? There is already a massive amount of data in this paper, so I do not expect the authors to perform new experiments. However, these higher potentials do accelerate the rate of Pt NP growth and agglomeration compared to the load cycle protocol (though the data is often convoluted by the high rates of carbon corrosion). Thus, it would also be useful if the authors discussed/hypothesized about the impact of performing this work at higher UPLs, particularly since the development of more corrosion stable carbon and non-carbon supports is a highly active area of study.

Author reply: We genuinely thank the reviewer for raising this insightful suggestion. Although we did not test the durability of these catalysts using higher UPLs in this work, we could expect that the PtNiMoAu catalyst would present the improved durability even under the higher UPLs compared to other reference catalysts. This can be ascribed to the stabilization effects of Mo and Au doping for PtNi-based catalysts, which would basically reduce the rate of Pt and Ni dissolution even under higher UPLs and thus suppress the sintering process. Besides, it is of great importance to strengthen the stability of the carbon support under higher UPLs, as the reviewer stated. We

have added the relevant discussion to the revised manuscript (page 3, line 15; page 12, lines 8-10).

6. *Durability Testing:* On page 14: the authors state that the upper potential limit (UPL) used in their durability tests is 1.0 V vs RHE. However, Table S4 states the UPL is 1.1 V vs RHE. One of these is incorrect (most likely the Table) and needs to be fixed.

Author reply: We genuinely thank the reviewer's careful reading. We have corrected the upper potential limit (UPL, 1.0 V vs RHE) used in their durability test (Supplementary Table 5).

Reviewer #3

The manuscript by Gao et al. reports on the study of PtNiAuMo alloys as the catalysts for the oxygen reduction for the application in fuel cells and clarifies the role of Au and Mo dopants on the enhancement of the activity and stability of the multimetallic alloy. The system described in the work demonstrates the performance surpassing that of the benchmark Pt/C and PtNi catalysts. The manuscript is well-structured and can be recommended for publication in Nature Communications once the following concerns are addressed:

Author reply: We genuinely thank this reviewer for his/her constructive and positive comments.

1. *The results of DFT calculations for ORR on Pt(111) discussed on P. 7-8 and Fig. S13 are significantly different from the published data on Pt(111) slab: in a seminal study [Phys. Chem. Chem. Phys., 2008, 10, 3722], the overpotential for ORR on Pt(111) is 0.75 eV which is significantly lower than the values reported in this manuscript not only for Pt, but also for PtNi, PtNiMo, PtNiAu. Referee acknowledges that different calculation parameters will have an impact on the absolute energy values, yet such an inconsistency for the relatively simple and well-studied system like metallic platinum has to be carefully addressed to support the validity of the DFT data in the manuscript.*

Author reply: We genuinely thank this reviewer for his/her constructive comment. In this revised manuscript, we adjusted the calculation parameters, i.e. using a larger supercell and smaller

convergence value for the energy and force, to improve the calculation accuracy. In this case, the calculated overpotential for ORR on Pt(111) is 0.73 eV, very close to the value (0.75 eV) reported in the mentioned reference (Phys. Chem. Chem. Phys., 2008, 10, 3722). Also, the trend in the overpotential for ORR on different slabs agrees well with the activity trend obtained in experiments (see Supplementary Figs. 10, 13, and 14a,b). We have revised the relevant figure and description in the revised manuscript (**page 16, lines 6, 14, 16, and 17**).

2. *What is the reason for a decrease of the Pt^{2+} fraction in PtNiMoAu alloy as compared to PtNi (Fig. 3g)? Is it because a smaller fraction of Pt sites is located on the surface of PtNiMoAu? This is also likely the dominant reason for the lower intensity of the white line in Pt L-edge XAS spectrum, rather than the electronic effects from the neighboring atoms as discussed on P. 9.*

Author reply: Because the atomic percents of Pt and Ni in PtNiMoAu and PtNi alloy NWs, as well as the diameters, are almost the same, we thus speculate that the fraction of Pt site on the surface of catalysts is very close. In this case, we can reasonably ascribe the decrease of Pt^{2+} fraction in PtNiMoAu NWs to the presence of more electron transfer from neighboring atoms to Pt.

3. *Analysis of the in situ XANES data indicating that oxidation of Pt in PtNiMoAu occurs at higher potential than for PtNi should be supplemented with the analysis of CV data, which should demonstrate the anodic shift of Pt/Pt^{2+} redox potential. These changes of the formal redox potentials of platinum should be explained in terms of the inductive effect from Mo/Au.*

Author reply: We genuinely thank this reviewer for his/her constructive comment. As requested, we have analyzed the CV of PtNiMoAu NWs and PtNi NWs, confirming the anodic shift of Pt/Pt^{2+} redox potential for the PtNiMoAu NWs compared to the PtNi NWs. We have added the relevant figure and description in the revised manuscript (Supplementary Fig. 25; **page 11, lines 1-2**).

4. *Fig. S16 is absolutely confusing and should be removed as it represents a fitting of the noise.*

Author reply: As suggested, we have removed the figure.

REVIEWER COMMENTS

Reviewer #1 (Remarks to the Author):

The authors made many efforts that indeed improve the manuscript. The reviewer appreciated it. However, there are still some technical issues to be addressed.

The experimental details of MEA tests should be applied, at least including the ionomer/carbon ratio in the catalyst ink, the type of GDL/MPL, and the compression ratio of GDL.

Considering that a single serpentine flow channels were used, the gap pressure drop between the inlet and outlet should be tested and the possible influence of gap pressure drop on the MEA test should be discussed.

The detailed test and calculation method of MA in MEA should be supplied.

The reviewer did not find the data of the Ni content of the catalysts before and after the MEA durability tests (Table S6 showed the change of metal content of the catalysts during the RDE instead of MEA tests). The data of metal content change of the catalysts after the MEA durability test is necessary; and the characterization details of metal contents of the catalysts upon MEA durability tests should be supplied (for example, STEM mapping, etc.).

The data of ECSA and Ni content change of the catalysts during the MEA durability test should be shown in the main text instead of SI.

It is better to replace of "PtNi alloy catalyst" with "PtNi nanowire catalyst" in the title, as the authors only demonstrated the nanowire catalysts in this work.

Reviewer #2 (Remarks to the Author):

I am satisfied with the revisions made by the authors

Reviewer #3 (Remarks to the Author):

The authors satisfactorily addressed my comments, the work can be recommended for publication.

A point-by-point response to reviewer comments

Manuscript ID: NCOMMS-23-28934A

Title: Identifying the distinct roles of dual dopants in stabilizing the PtNi alloy catalyst for durable fuel cell

Reviewer #1

The authors made many efforts that indeed improve the manuscript. The reviewer appreciated it. However, there are still some technical issues to be addressed.

Author reply: We genuinely thank this reviewer for his/her constructive comments, which indeed improve the manuscript greatly.

1. The experimental details of MEA tests should be applied, at least including the ionomer/carbon ratio in the catalyst ink, the type of GDL/MPL, and the compression ratio of GDL.

Author reply: As requested, we have provided the detail information on MEA tests. Specifically, the ionomer/carbon ratio of 0.8 in the catalyst ink, the GDL of Toray TGP-H-060, and 31.5% compression ratio for GDL were adopted. The relevant contents have been added to our revised manuscript (**page 15, lines 15-16 and 18-19**).

2. Considering that a single serpentine flow channels were used, the gap pressure drop between the inlet and outlet should be tested and the possible influence of gap pressure drop on the MEA test should be discussed.

Author reply: We genuinely thank this reviewer for his/her constructive comments. We have tested the pressure of the inlet and outlet at different flow rates and calculated the corresponding pressure drop, as shown in Fig. S26. Specifically, the gap pressure drop between the inlet and outlet for anode (H_2 , 200 mL min^{-1}) and cathode (O_2 , 500 mL min^{-1}) is 3 kPa and 6 kPa, respectively. Given the relatively low pressure drop, we reasonably believe that it has negligible

effect on MEA performance in this work. The relevant content has been added to our revised manuscript (**page 11, lines 15-19; page 15, lines 25-29**).

3. The detailed test and calculation method of MA in MEA should be supplied.

Author reply: As requested, we have added the detailed test and calculation method of MA in MEA. Specifically, the MA was calculated by normalizing the measuring currents with Pt loading amount in 150 kPa_{abs} H₂/O₂ (80 °C, 100% RH, 200_{anode}/500_{cathode} sccm) at 0.9 V_{iR-free}. The relevant content has been added to our revised manuscript (**page 15, line 15; page 16, lines 1-2**).

4. The reviewer did not find the data of the Ni content of the catalysts before and after the MEA durability tests (Table S6 showed the change of metal content of the catalysts during the RDE instead of MEA tests). The data of metal content change of the catalysts after the MEA durability test is necessary; and the characterization details of metal contents of the catalysts upon MEA durability tests should be supplied (for example, STEM mapping, etc.).

Author reply: We genuinely thank this reviewer for his/her careful reading of our manuscript. In fact, we had tested the Ni content of the catalysts by ICP-MS before and after the MEA durability tests, as shown in Table S6. However, due to our handwriting error, the RDE in the legend of Table S6 should be MEA. Thus, we have corrected the legend in the Table S6. Besides, after MEA durability tests, the smallest change of morphology for PtNiMoAu NWs/C in all NWs/C catalysts also supports changing trend of Ni content (Fig. S31). The relevant content has been added to our revised manuscript (**page 12, line 12**).

5. The data of ECSA and Ni content change of the catalysts during the MEA durability test should be shown in the main text instead of SI.

Author reply: As suggested, we have plotted the graphs for the ECSA and Ni content change of the catalysts during the MEA durability test, as shown in Fig. 6d,e.

6. It is better to replace of “PtNi alloy catalyst” with “PtNi nanowire catalyst” in the title, as the authors only demonstrated the nanowire catalysts in this work.

Author reply: We genuinely thank this reviewer for his/her constructive suggestion. “PtNi alloy catalyst” has been replaced with “PtNi nanowire catalyst” in the title.